# Exploring the role of early-life circumstances, abilities and achievements on well-being at age 50 years: evidence from the 1958 British birth cohort study

Brian Dodgeon [ORCID],[1] Praveetha Patalay,[1,2] George B Ploubidis,[1] Richard D Wiggins[1]

[1]Centre for Longitudinal Studies, University College London Social Research Institute, London, UK
[2]MRC Unit for Lifelong Health and Ageing, University College, London, UK

**Correspondence to**
Brian Dodgeon;
b.dodgeon@ucl.ac.uk

## ABSTRACT

**Objectives** We aim to examine the relative contributions of pathways from middle childhood/adolescence to mid-life well-being, health and cognition, in the context of family socio-economic status (SES) at birth, educational achievement and early-adulthood SES. Our approach is largely exploratory, suspecting that the strongest mediators between childhood circumstances and mid-life physical and emotional well-being may be cognitive performance during school years, material and behavioural difficulties, and educational achievement. We also explore whether the effects of childhood circumstances on mid-life physical and emotional well-being differ between men and women.

**Setting/participants** Data were from the National Child Development Study, a fully-representative British birth cohort sample of 17 415 people born in 1 week in 1958.

**Primary/secondary outcome measures** Our four primary mid-life outcome measures are: cognitive performance, physical and emotional well-being and quality of life. Our intermediate adult outcomes are early-adulthood social class and educational/vocational qualifications.

**Results** Using structural equation modelling, we explore numerous pathways through childhood and early adulthood which are significantly linked to our outcomes. We specifically examine the mediating effects of the following: cognitive ability at ages 7, 11 and 16 years; childhood psychological issues; family material difficulties at age 7 years: housing, unemployment, finance; educational/vocational qualifications and social class position at age 42 years.

We find that social class at birth has a strong indirect effect on the age 50 outcomes via its influence on cognitive performance in childhood and adolescence, educational attainment and mid-life social class position, together with small direct effects on qualifications and social class position at age 42 years. Teenage cognitive performance has a strong positive effect on later physical health for women, while educational/vocational qualifications have a stronger positive effect on emotional well-being for men.

**Conclusion** Our findings provide an understanding of the legacy of early life on multiple aspects of mid-life health, well-being, cognition and quality of life, showing

## Strengths and limitations of this study

► The richness of its longitudinal dimension, conveying the importance of early-life circumstances on early adult and mid-life measures of achievement, health and well-being.

► The integrated nature of the analysis simultaneously models pathways to four mid-life outcomes, three of which are strongly correlated after the inclusion of several key early-life antecedents.

► It makes optimal use of all the available longitudinal data by employing full information maximum likelihood estimation to take account of item non-response.

► It shows the enduring importance of parental social class for cognitive performance and later life outcomes, controlling for family material difficulties in childhood.

► A potential limitation is that our model does not include a broader set of demographical and psychosocial factors: a result of our decision to restrict the number of early-life influences to avoid over-complexity in the statistical modelling. There is also a small amount of attrition bias between birth and age 50 years.

stronger mediated links for men from childhood social class position to early adult social class position. The observed effect of qualifications supports those arguing that education is positively associated with subsequent cognitive functioning.

## INTRODUCTION

Authors since the 1990s have advanced the importance of a life course approach to explore the relationship between early-life circumstances and later life health outcomes.[1 2] Carr[3] emphasises how diverse early-life experiences affect physical, emotional and cognitive health in later life by drawing on a wealth of longitudinal data across USA, Australia and Europe. Our

empirical study is based on a single British birth cohort which covers several waves of data collection from birth (1958), middle childhood (age 7 and 11 years), adolescence (age 16 years) and mid-life at ages 42 and 50 years for over 8000 sample members. Age 50 years represents a specific midpoint of mid-life (40–60 years) as defined by Midlife Development in the United States: Brim *et al*[4] suggest 'Midlife has been described as the last uncharted territory of the life course'. To this extent our conceptual framework is largely informed by Elder[5]: it may be described as a cumulative (dis-) advantage model[6] where we examine the legacy of early-life circumstances, notably the British Registrar General's (RG) classification of occupations (RG social class)[7] as having lasting effects on health and well-being at age 50 years. Our approach implies a reciprocal relationship between socio-economic status (SES) and health, and allows for the possibility that favourable circumstances in later life can act counter to the effects of earlier disadvantage.[8]

To this end we select outcomes at age 50 years covering four distinct areas: self-assessed quality of life (QoL), physical and emotional well-being (PWB and EWB, respectively), and a cognitive ability test.

Our research interest is in exploring the extent to which these outcomes inter-relate, adopting an empirical view that, taken together, these measures would tell us more about the consequences of the past than studied individually.[9] There is a growing body of evidence based on UK longitudinal studies[10–12] that provides a strong argument for looking at these outcomes together within the same model. Cooper *et al*[13] provided both substantive and empirical evidence for examining gender differences in this analysis.

Drawn from the same rich longitudinal resource, our predictors and mediators begin with birth circumstances: birth weight, breastfeeding, maternal smoking during pregnancy and parental SES. From the age 7 years sweep of the study we include the effects of family material difficulties such as unemployment, finance and housing, in conjunction with childhood scores of socio-emotional adjustment and cognitive performance, each measured on three occasions spanning the ages 7, 11 and 16 years. The inclusion of highest qualification and SES at age 42 years in the model enables us to trace which childhood effects persist, and which are built on or attenuated by education and social mobility.

Our analysis builds on the framework adopted by Wood *et al*[14] on the prediction of mental well-being as a single outcome[15] across four UK birth cohort studies, which showed that childhood SES is directly and indirectly (through adult socio-economic pathways) linked to adult EWB. We expand on their approach by including four outcome measures, although for a single birth cohort.

Our analysis adopts a structural equation modelling (SEM) approach,[16] in order to explore the pathways from childhood and adolescence to our multivariate mid-life outcomes for both men and women.

## DATA AND METHODS

Our data source, the British National Child Development Study is fully representative of the population of Great Britain, having interviewed the parents of 98.5% of all babies born in the same week of 1958,[17] and followed them up longitudinally with 10 subsequent interviews throughout life. It provides operational measures regarding birth circumstances, the early and teen years covering ages 7, 11 and 16 years, and adulthood at ages 42 and 50 years, respectively.

The analysis is based on 3815 male and 4209 female cohort members present at each occasion who all have at least one genuine response for each of the four separate age 50 life domains, and who also participated in at least one of the three cognitive and socio-emotional assessments, in order to permit a reliable strategy for handling missing items.

We are aware there is a certain amount of attrition bias as a result of differential loss-to follow-up from the 17 415 cohort members present at the birth study in 1958. As a check on the impact of sample loss, we were reassured by the close match of the distribution of the birth social class variable within our sample of 8024 against that of the same variable within the full 17 415 birth sample ($\chi^2$=1.29, 5 df, non-significant) (see online supplementary appendix 1).

Our outcomes at age 50 years cover four distinct measures, the first two being self-reported PWB and EWB measured using subscales[18] of the RAND 36-Item Short-Form Health Survey[19–23]. Third, a self-reported measure of QoL covering the life domains of Control, Autonomy, Self-Realisation and Pleasure (CASP-12 v2)[24]: a theoretically-informed measure of subjective well-being, closely following the approach adopted by Diener and Suh.[25] The fourth domain outcome, cognitive ability (hereafter Age 50 Cog) consists of four cognitive tests examining memory (word recall and delayed recall), executive function (animal naming), attention and mental speed (letter cancellation).[26]

Online supplementary appendix 2 lists all of the manifest and latent variables employed in the analysis. Indicators for cognitive function at age 50 years are listed here, while those for physical, emotional and QoL outcomes are expanded on more fully in online supplementary appendix 3. Childhood behavioural indicators are listed in online supplementary appendix 4 following Rutter *et al*.[27] We recognise of course that Rutter scores may be subdivided into constituent dimensions of problem behaviour (withdrawn/anxious/oppositional/inattentive),[28] but for this paper we prefer to use Rutter total scores, following Parsons *et al*.[29]

In fitting our structural equation models using the MPlus software[30] we obtain direct and indirect (mediated) effects of our background variables in terms of childhood social origins and social adjustment, on outcomes in early adulthood and mid-life at age 50 years. We regard all four of our birth indicators as primary predictors, valuing the opportunity to assess the relative impact of each for policy interventions.[31] Our birth

social class variable is based on father's occupation, since in 1958 when the cohort were born, only 38% of mothers were economically active.

The sets of predictors determining our paths differ slightly for each of the four outcomes so as to avoid the problem of 'seemingly unrelated regression'.[32]

We employ full information maximum likelihood estimation.[33] This addresses the issue of item non-response arising where questions are avoided or not answered by certain participants.

The SEM procedure that follows begins with an analysis of the sample as a whole, followed by separate analyses for men and women, supported by a multigroup analysis[34] to test for the structural equivalence of the models for men and women. Model 'goodness of fit' is assessed via a combination of conventional criteria: root mean square error (RMSEA), comparative fit index, (CFI) and Tucker Lewis fit index, (TLI) where RMSEA<0.05, CFI>0.90 and TLI >0.90 would typically provide evidence of 'good fit,' though each of these criteria individually are regarded as indicative, and are not always strictly adhered to as arbiters of model validity.[35]

Having a large number of SEM pathways involving 261 free parameters, we had regard to the n:q rule[36] and were satisfied that our sample-size-to-parameters ratio was comfortably within Jackson's guideline of 10:1. Our application of SEM involved the use of modification indices, which led to our running correlations on suggested variables. All our final models were identified, all manifest variables loading onto the respective latent variables with satisfactory fit. The details of the individual assessments of our measurement models appear in online supplementary appendix 5.

### Patient and public involvement

Participant consent was obtained from parents of cohort members at the initiation of the original birth cohort study in 1958. On attaining adulthood, permission was obtained from cohort members themselves at each subsequent longitudinal sweep.

### RESULTS

The results begin with consideration of the inter-correlations between our mid-life outcome measures. Further univariate descriptives are contained in online supplementary appendix 2. Figure 1 contains a final SEM diagram for the sample as a whole. Subsequent reference will be made to online supplementary appendix 6 for the details of mediation, total and indirect effects. Online supplementary appendix 7 contains separate analyses by sex, and online supplementary appendix 8 contains further information on our multigroup analysis based on gender.

Note that all coefficients reported in the Results section are standardised.

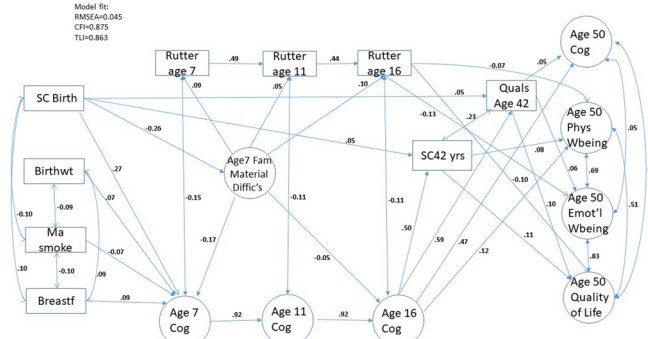

**Figure 1** Estimated pathways of the structural equation model, N=8024 (showing paths significant at p<0.001).

### Descriptives and correlations

As a preliminary inspection of the inter-relationships between our four outcome measures, table 1 contains a correlation matrix of the four summative indices based on these measures for both men and women.

Figure 1 contains estimates of association among these outcomes as defined by measurement models under a full SEM analysis. As a check on the robustness of our statistical findings the SEM analyses were subject to bootstrap analysis under 1000 repetitions. We also carried out a check on the influence of outliers (results available on request).

### Birth social class effects on early mid-life outcomes

The path diagram shown in figure 1 provides the model results for the sample as a whole showing only statistically significant paths at a p-level of 0.001, employing a Bonferroni correction for the 50 path coefficients being estimated (0.05/50=0.001).[37]

While there are various pathways that connect early-life circumstances with cognitive performance, family material difficulties and social adjustment, there remains a small direct effect of social class at birth on each of our two early adult outcomes: Age 42 social class and educational/vocational qualifications (0.05 for both paths). These two 'interim' or 'pivotal' destinations, in turn, have notable effects on our four mid-life outcomes at age 50 years, which we describe in the section Indirect effects via adulthood mediators.

### Indirect effects via pathways through childhood mediators

Inspecting the paths in figure 1 from birth circumstances through ages 7, 11 and 16 years, we see first, that all four of our birth characteristics have direct effects on cognitive performance at age 7 years, when children are in primary school. Social class at birth has the most notable association with age 7 cognition (0.27). This cognitive variable in turn has a strong direct effect on age 11 cognition (0.92) and similarly for age 11 years on cognitive performance at age 16 years (0.92).

There is a clear path via cognitive ability throughout childhood and adolescence, to adult achievement as indicated by Age 42 social class and educational qualifications

**Table 1** Associations between the four sets of mid-life outcomes as summative indices (men/women)

| Age 50 domain | Age 50 Cog | PWB | EWB | QoL |
|---|---|---|---|---|
| Age 50 Cog | 1.00/1.00 | | | |
| PWB | 0.12/0.15 | 1.00/1.00 | | |
| EWB | 0.12/0.09 | 0.57/0.58 | 1.00/1.00 | |
| QoL | 0.15/0.10 | 0.48/0.48 | 0.63/0.64 | 1.00/1.00 |

Interestingly, QoL, PWB and EWB demonstrate moderate associations of inter-correlation. While cognitive ability is positively related to the PWB and EWB measures, the strength of the association is modest, with PWB having a higher coefficient for women, and EWB and QoL a higher coefficient for men.

Age 50 Cog, cognitive ability; EWB, emotional well-being; PWB, physical well-being; QoL, quality of life.

(Age 16 Cog to social class 42 years 0.50, Age 16 Cog to Quals 0.59).

This legacy of childhood cognitive performance is unsurprising, but interestingly Age 16 Cog also has a direct association with age 50 PWB (0.12) besides more naturally, Age 50 Cog (0.47).

Family material difficulties in childhood, themselves associated with birth social class (−0.26), also play their part in understanding a child's trajectory in cognitive performance. Characterised by issues with housing, finance and employment when a child is 7 years old, these family material difficulties are seen to have negative consequences for cognitive performance at age 7 years (−0.17) and a lagged negative effect on later performance at age 16 years (−0.05).

Additional analyses also show that family material difficulties impact on a child's emotional and behavioural scores (based on Rutter) at ages 7, 11 and 16 years (0.09, 0.05 and 0.10, respectively). In turn these scores had small significant negative correlations with cognitive performance at each corresponding age (−0.15, −0.11 and −0.11, respectively). But most notably, Rutter scores at these three ages are significantly correlated (0.49 for age 7–11 years and 0.44 for age 11–16 years), and there are direct effects from age 16 Rutter scores to three of our four age 50 outcomes: EWB (−0.13), QoL (−0.10) and PWB (−0.07).

### Indirect effects via adulthood mediators

Adult social class and lifetime qualifications are, together, influential predictors of each of our four mid-life outcomes: adult social class has significant predictive effects (0.08 and 0.11) on PWB and QoL.

In general, the socially-advantaged enjoy better health and report higher levels of QoL than their less advantaged counterparts. Interestingly, qualifications predict Age 50 Cog ability (0.05), EWB (0.06) and QoL (0.10) but not PWB.

### Gender differences

Our multigroup analysis reveals that certain paths have significantly different effects for men and women. In online supplementary appendix 7 we display two separate path diagrams for men and women. For men, there is a direct effect of social class at birth on Age 42 social class

and qualifications (0.09 and 0.07, respectively), but for women there is no significant effect.

Both sexes see the birth social class effect remains equally strong for cognitive performance at age 7 years and beyond, but women experience a slightly worse negative effect of social disadvantage at birth on the latent variable 'family material difficulties' (−0.27 compared with −0.24 for men). In turn, family material difficulties have slightly more influence on age 7 cognition (−0.18) for men than for women (−0.16), and a small lagged effect on Age 16 Cog only evident for women (−0.05). The Rutter social adjustment scores at age 11 years do not significantly correlate with the cognitive performance scores for men, but they do for women at ages 7, 11 and 16 years (−0.15, −0.11 and −0.13 compared with men's −0.14 at age 7 years, −0.11 at age 16 years).

Early adult social class has a slightly stronger influence for men than women on our age 50 outcomes PWB and QoL, and qualifications have a significant effect on EWB for men, but not for women. In contrast, women's Age 16 Cog has a significant effect on their later PWB (0.17), an effect not significant for men; and the path from Age 16 Cog to Age 50 Cog has a stronger coefficient for women (0.49) than men (0.43).

We pursue the evidence for gender differences via a sensitivity analysis under MPlus which formally tests for statistical difference between paths for each sex. Online supplementary appendix 8 identifies six paths where the effects for men and women are significantly different, but only three have notable differences in terms of their effect sizes in the multigroup analysis. These are:

► Stronger effect for women from Age 16 Cog to Age 50 Cog as noted above (0.49–0.43).
► Direct effect of social class at birth on Age 42 social class (0.09 for men, no significant effect for women).
► Link for women from Age 16 Cog to PWB (0.17) (no significant effect for men).

### DISCUSSION

SEM has elucidated the nuances of the influence of pathways from childhood and early adulthood to our four related outcomes of health, well-being and cognition in mid-life. The work demonstrates the benefits of an integrated approach to defining the aspects of the life course

that affect well-being in mid-life. Physical and emotional health are important factors affecting a person's subjective account of their QoL, and PWB correlates at 0.69 with EWB, which in turn correlates 0.83 with QoL. While cognitive performance at age 50 years is relatively weakly correlated, the other three are closely intertwined. These associations are similar in magnitude to those reported in table 1; however, they are now estimated by taking measurement error and the legacy of the life course into account.

We also show that some of the paths to health and well-being in mid-life are different for men and women.[38] However, important communalities remain.

Social origins, in particular social class at birth, have a major indirect influence on mid-life outcomes, the mediation process beginning with a child's cognitive performance throughout formal schooling and subsequently via cognitive performance at age 16 years. Collectively, these influences are weakened or reinforced by the existence of family material difficulties in early childhood (themselves influenced by childhood social class). While cognitive performance is a strong predictor of early adult social class and educational status for both men and women, social class at birth also has a lasting association with these mid-life indicators for men but not for women.

This could well suggest that women have a looser connection with the legacy of social class at birth, perhaps partly because their path to advantaged SES at age 42 years may be impeded by caring responsibilities and gender inequalities in the labour market.[39]

Early adult social class influences all four of our mid-life outcomes, but we see the direct effect of social class at birth on social class and qualifications in early adulthood is pronounced for men (0.09 and 0.07, respectively), but not significant for women.

A relatively small but consistent direct effect of qualifications on Age 50 Cog supports those arguing that education is positively associated with subsequent cognitive functioning,[40] although we are aware that there are counter-arguments to this view[41]. Equally, the relationship between QoL and early adult social class finds support in the work of Blane et al.[42]

The strength in the evidence presented here lies in the richness of its longitudinal dimension which conveys the importance of early-life circumstances on early adult and mid-life measures of achievement, health and well-being.

One limitation of our study is that our model does not fully take into account all of the potential demographical and psychosocial factors other researchers have studied as predictors of satisfactory mid-life development.[28] Nevertheless, it can be argued that our birth social class variable acts as a 'proxy' for certain parental beliefs and behaviours.[43] We are embarking on developing a conceptual framework and modelling strategy to extend our empirical analyses in the future as our cohort members age: in particular, our subjects are not yet old enough for us to assess whether any decline in cognitive performance at age 50 years will lead to subsequent decline and

become more influential on EWB and QoL. However, we were able to show that cognitive performance in the teen years holds a legacy for cognitive performance in mid-life. A final limitation is a certain amount of attrition bias between birth and age 50 years (differential non-response) which was addressed earlier in the paper and in online supplementary appendix 1.

Social class at birth has a strong influence on early cognitive performance and beyond, which in turn has consequences for adult achievement and social class. Clearly, the presence of family material difficulties and behavioural issues will impede (mediate) this trajectory. Interestingly, childhood behaviour, in particular at age 16 years, has a lasting effect on mid-life PWB, EWB and QoL.

In sum, health, emotion, QoL and to a lesser extent cognitive function do not stand alone as markers of well-being in mid-life. They represent the constituents of an inter-related whole which is shaped by early-life circumstances, family material difficulties, social adjustment and cognitive performance in childhood.

**Acknowledgements** We would like to acknowledge the thoughtful and constructive comments of our three reviewers, which strengthened our paper.

**Contributors** BD performed the data management and structural equation modelling using the Mplus software, and took responsibility for completing all aspects of the paper. PP contributed to the conceptual planning of the analysis jointly with RDW, performed preliminary regression modelling and advised on aspects of the structural equation modelling (SEM) strategy. GBP advised on many aspects of the statistical analysis, and provided detailed comments on the drafting of the paper. RDW undertook the conceptual planning of the analysis jointly with PP, and initiated the SEM approach, continuing to provide advice on the models as they became more complex, and collaborating with BD on the write-up of the findings.

**Funding** Assistance is acknowledged from ESRC grants ES/M001660/1, ES/N003683/1 and ES/M008584/1.

**Competing interests** None declared.

**Patient consent for publication** Not required.

**Ethics approval** Ethical approval was not required, since the study used the secondary data of the National Child Development Study, which itself had received ethics consent from the London Research Ethics Committee, ref 08/H0718/29 for its age 50 survey, the source of our outcome data. Multicentre Research Ethics Committee (MREC) approval was received for earlier NCDS surveys.

**Provenance and peer review** Not commissioned; externally peer reviewed.

**Data availability statement** Data may be obtained from a third party and are not publicly available. The analysis is based on secondary data available for download from the UK Data Archive at https://beta.ukdataservice.ac.uk/datacatalogue/series/series?id=2000032. Syntax used to derive variables for this analysis can be obtained by e-mailing the corresponding author, Brian Dodgeon.

**ORCID iD**
Brian Dodgeon http://orcid.org/0000-0002-7130-8761

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
