## [Reviewer comments · BMJ Open]

ARTICLE DETAILS

TITLE (PROVISIONAL)	Exploring the role of early life circumstances, abilities and achievements on well-being at age 50 years: evidence from the 1958 British Birth Cohort Study
AUTHORS	Dodgeon, Brian; Patalay, Praveetha; Ploubidis, George; Wiggins, Richard

VERSION 1 – REVIEW

REVIEWER	Erin Barker Department of Psychology Concordia University Montreal, Quebec, Canada
REVIEW RETURNED	14-Jun-2019

GENERAL COMMENTS	The authors of the current study use a large birth cohort sample to examine the effects of social status from birth to mid-life on mid-life functioning. Furthermore, they examine mediated pathways across later child and adolescence via cognitive functioning and behavior problems. The study has great potential, but additional conceptual refinement of the model at both ends of the life course – childhood/adolescence and midlife – is needed to maximize the contribution to the literature. The overall presentation of the methods and results could also be improved. Below, I pose several questions and make several comments that if addressed will improve the quality of the research. With respect to the childhood/adolescence part of the conceptual model: 1. Why were the birth health indicators (smoking, birth weight, breast feeding) included in the model as separate manifest variables rather than combined and included as a single latent variable? Given that the main variable of interest across time is social class and that the patterns of association are similar for these three variables, combining these variables and conceptualizing their inclusion in the model as “controlling” for their effects would simplify the model and emphasize the goal of examining the effects of social class from birth through midlife on midlife outcomes.2. Why were the Rutter measures included as manifest total scores/sums rather than latent variables representing different dimensions of problem behavior? There is a large body of research that grew from the development of this measure that shows that this measure taps different dimensions of child problems (e.g., withdrawn/anxious behavior, oppositional behavior, inattentive behavior). And, in that large developmental psychopathology literature differential associations between the subtypes of behavior
--

problems with cognitive functioning and family functioning have been identified. It has also been found that the age at which different types of behavior problems are presented matters for later outcomes (i.e., childhood onset vs. adolescence onset). It is important to model these nuances because the patterns of behavior may have different consequences for later developmental outcomes. The large sample size affords the authors the opportunity to incorporate this level of detail/nuance into their model and doing so would greatly increase the contribution the study makes to the literature. I refer the authors to the work of R. McGee, S. Hinshaw, A. Caspi and T. Moffitt, and L. Serbin for some examples of what I have described above.

3. With respect of the family difficulties measure at age 7, I am wondering why it is not aligned with the other age 7 measures in Figure 2. It would be easier to follow the developmental timing of the measurement waves if all variables measured at a given time point were aligned vertically. More substantively, with respect to this measure, a more precise label is needed. "Family difficulties" could mean so many things – parenting challenges, stressful events, family discord, etc. But, the difficulties measured in the current study are specifically related to financial difficulties – housing, employment, finances. Thus, the variable should be labelled as "Family financial difficulties" or something similar.

With respect to the midlife part of the model:

1. It is unclear why and how the age 50 measures made. This is due in part to the fact that the tables describing the measures in the appendices are challenging to follow. A lot of descriptive information is repeated (e.g., labels, scale anchors) that could be presented more concisely. But, I am particularly concerned by the fact that social functioning and emotional functioning were combined into one "Emotional Well-Being" variable. What is the rationale for this? Typically, social functioning (e.g., social support network size, quality, etc.) is conceptualized as a protective factor that supports well-being, both physical and emotional. I think these should be included as separate constructs in the model. Additionally, the measure of quality of life appear to be more of a subjective well-being measure, akin to the Ryff Scales. Given its strong relationship with the emotional well-being measure, I question whether both should be included in the model or if one is a sufficient indicator of this aspect of well-being at age 50. Here I refer the authors to the work of L. Carstensen and S. Charles, and C. Ryff.

Minor points that require clarification:

1. In the SEM model, it would be helpful if there was consistency in the labelling with respect to age. That is, label all variables with their age label, like age50cog. Or align all variables vertically by age and add labels above or below the figure.

2. As previously mentioned, the tables are difficult to follow and more precise labelling of columns and rows is needed. For example, it is unclear what is presented in the column labelled "coding." Is this the possible or actual range? Seems to be the actual range, and it should be labelled as such.

3. It is unclear what the social class categories represent other than 1 = low. Do they map on to income brackets or some other meaningful metric that would help the reader understand the

	ranking? The same point applies to the qualifications measure. 4. Throughout the paper, there is an overuse of acronyms without sufficient description/definition, including in the figure captions.
--	--

REVIEWER	Kaitlin P. Ward, MSW University of Michigan, USA
REVIEW RETURNED	30-Jun-2019

GENERAL COMMENTS	Pathways from early childhood to physical & emotional well-being, quality of life and cognitive function in mid-life This manuscript examines the associations between multiple circumstances present at birth; behavioral, emotional, and socio-economic constructs in middle childhood and adolescence; and adult outcomes. The manuscript primarily focuses on mediating mechanisms and aims to help explain mid-life outcomes. Strengths of the paper include a large sample size and the utilization of SEM. Weaknesses include insufficient study conceptualization, insufficient descriptions of methods and data analysis, and a general lack of purpose (in both statistical choices and study conceptualization). Specific recommendations can be found below. Introduction 1. In the abstract and the introduction, it is unclear what the predictors in the model are, and why the authors have focused on these specific predictors. The introduction should lead the reader to understand why you chose the specific predictors you did. The authors keep mentioning childhood SES in the introduction, leading the reader to assume this would be the main focus. However, the it seems the authors are focusing on SES, birthweight, maternal smoking, and breastfeeding? Or, is social class the authors' main predictor variable and birthweight, maternal smoking, and breastfeeding are covariates? I find myself constantly re-reading the article to figure out what the main predictor is. a. In relation to the above point, the abstract mentions "childhood circumstances" broadly, then mentions birth SES and teenage cognition. The introduction should lead the reader to understand why you chose what time points you did (i.e., early childhood or adolescence). b. To be clear, the authors have centered the article—even the article title—around the idea that they are focusing on early childhood. From what I understand, the main predictor that the authors focus on is the father's social class at the child's birth. That's not early childhood. There is also birthweight, maternal smoking and breastfeeding, but this is not early childhood either. These are measures at the time of birth. Then, the authors have measures at 7, 11, and 16. This is not early childhood; this is middle childhood to adolescence. It seems to me that they authors do not even have a single measure in the "early childhood" period. 2. In the introduction, the authors mention various theories that they are actually not testing. For example, the authors are not testing cumulative advantage/disadvantage; they are not testing for compensation; they are simply testing mediation. Instead of spending time discussing theories that the authors cannot and do not test in their analyses, the authors should be presenting theories (or at least justification) or WHY they are testing what they're testing. Why SES? What theories about SES—and the mechanisms through which SES works to influence future outcomes—apply to your study
--

(perhaps Dr. Pam Davis-Kean's work would be helpful? Also Dr. Vonnie McLoyd and Dr. Rand Conger).

a. In relation to the point above, and as another example: the authors spend an entire paragraph discussing the mediating mechanisms that they never actually test. i.e., They don't measure whether parents are facilitating social/cultural/educational advantage. The authors need to tell the readers WHY the mechanisms that they actually test in the paper are important.

3. As the introduction stands, I am left wondering what the purpose of this study is. Why is the link from childhood SES to behavior in childhood/adolescence to age 50 outcomes important? Why is the link between childhood SES to cognition in childhood/adolescence age 50 outcomes important? Why do we care about those four outcomes specifically? As this introduction stands, I am not convinced that the variables/relationships the authors tested provide any meaningful contributions, academic or otherwise.

4. Why do the authors find the need to mention they examined fifty pathways? This makes me extremely concerned about issues of multicollinearity, model identification, whether the authors followed the n:q rule of SEM, whether the model was identified, and so forth. Additionally, why were fifty pathways necessary? The authors did not tell the reader in the introduction why it was important for them to test those pathways theoretically or practically.

5. Related to the conceptualization issues above, the authors need to provide a very convincing argument for why they would suspect that these pathways work differently for men and women. If the authors ran a multiple-group analysis, I should have a full paragraph in the introduction giving me an overview of the literature that would suggest that these mechanisms work differently for men and women.

a. Why would the authors hypothesize that the mediation effects would be stronger for men than women? There is very little available evidence in the literature to justify formulating such a hypothesis.

6. The authors need to be a lot clearer about what they're actually testing. The authors point to appendix 1, but I'm still left wondering why they chose the variables that they did. I'd like things to be very clear: "we wanted to know if cognitive ability, behavioral problems, and education in middle childhood and adolescence mediated the relationship between SES at birth and age-50 outcomes."

Methods/Results

1. The authors need to discuss their measures. I understand the authors have limited space, but they at least need to discuss the primary predictor variable(s) (which I'm still unsure as to what that is/they are). Separate sections that discuss the predictors, mediators, and outcomes would be great.

2. The authors need to discuss what the makeup of the sample looks like. Is it a truly representative sample? Is it mostly upper SES? Is it racially/economically diverse?

3. The authors need to provide alphas for all the measures with multiple indicators/items in the method section.

4. The authors need to present results of their measurement models. I at least need to know that they indicators all loaded onto the latent variable and that they had sufficient fit.

5. The authors need to mention how they made the model fit. Did they have to use modification indices? Did any of the indicators need to be correlated?

6. The authors need to briefly mention whether their analyses met statistical assumptions. Were there outliers, non-normality,

	multicollinearity issues? Did you transform any of your variables to meet assumptions? 7. In this study, did the authors have to use sampling weights or account for neighborhood/city clustering? Was this a multilevel model? 8. If the authors are using FIML, why did they restrict the sample to the specifications found on page 5 lines 9-16? 9. What was the amount of missing data on your primary predictors, mediators, and outcome measures? 10. Did the authors use bootstrapping for the mediation analysis? If so, how many? 11. Throughout the results section, are the authors reporting standardized results (betas)? 12. When the authors are describing the results for men and women, how did they determine what was “stronger” for each? Did the authors use the DIFFTEST function in Mplus? Was it simply testing for structural invariance? If it’s the latter, how did you test for structural invariance, and did the authors use the scaling correction factor as an adjustment for when Mplus uses MLR? 13. Why were there so few control variables? How did you settle on the control variables that you did? 14. Why are you focusing on fathers' social class as your main predictor? Most studies that examine SES utilize some sort of combination of mothers' attributes (e.g., maternal education and income). The authors would need to justify this focus on fathers, and mention this in the introduction as well. Discussion 1. In general, I’m left confused as to how this study contributes to what already exists in the literature; how it relates to theory; and what the reader should take away from this study. I’m not sure why the authors ran the analyses they did. 2. The language used throughout the discussion adds to the abovementioned confusion. What were we trying to “tease out?” What to the authors mean by physical and emotional health “matter?” 3. The authors' analyses do not support causation; the authors should remove all references to causation. 4. The authors fail to provide an adequate overview of the study's limitations in the discussion section.
--	---

VERSION 1 – AUTHOR RESPONSE

Reviewer: 1
Reviewer Name: Erin Barker

With respect to the childhood/adolescence part of the conceptual model:

1. Why were the birth health indicators (smoking, birth weight, breast feeding) included in the model as separate manifest variables rather than combined and included as a single latent variable? Given that the main variable of interest across time is social class and that the patterns of association are similar for these three variables, combining these variables and conceptualizing their inclusion in the model as “controlling” for their effects would simplify the model and emphasize the goal of examining the effects of social class from birth through midlife on midlife outcomes.

The legacy of social class at birth for later life is a primary concern of the paper. Having distinct

indicators of mother's health behaviour at birth provides an opportunity to assess the relative impact of these indicators for policy interventions. [Cable et al 2012, Sacker et al 2013, Power et al 1997].

2. Why were the Rutter measures included as manifest total scores/sums rather than latent variables representing different dimensions of problem behavior? There is a large body of research that grew from the development of this measure that shows that this measure taps different dimensions of child problems (e.g., withdrawn/anxious behavior, oppositional behavior, inattentive behavior). And, in that large developmental psychopathology literature differential associations between the subtypes of behavior problems with cognitive functioning and family functioning have been identified. It has also been found that the age at which different types of behavior problems are presented matters for later outcomes (i.e., childhood onset vs. adolescence onset). It is important to model these nuances because the patterns of behavior may have different consequences for later developmental outcomes. The large sample size affords the authors the opportunity to incorporate this level of detail/nuance into their model and doing so would greatly increase the contribution the study makes to the literature. I refer the authors to the work of R. McGee, S. Hinshaw, A. Caspi and T. Moffit, and L. Serbin for some examples of what I have described above.

The observations regarding various aspects of child behaviour, onset and functioning are well taken. We very much appreciate your reference to the work of Caspi et al 1995 in elucidating the origins of child and adolescent behaviour problems up to age 15, and have now included a reference to this work in our re-draft on limitations. We decided not to fully expand the dimensions of problem behaviour at this stage in our research, using the Rutter total scores as in Parsons et al 2017. However, from the vantage point of this paper we intend to use the opportunity to incorporate further analyses which expand these dimensions in more detail.

3. With respect of the family difficulties measure at age 7, I am wondering why it is not aligned with the other age 7 measures in Figure 2. It would be easier to follow the developmental timing of the measurement waves if all variables measured at a given time point were aligned vertically. More substantively, with respect to this measure, a more precise label is needed. "Family difficulties" could mean so many things – parenting challenges, stressful events, family discord, etc. But, the difficulties measured in the current study are specifically related to financial difficulties – housing, employment, finances. Thus, the variable should be labelled as "Family financial difficulties" or something similar.

We have re-labelled the measure 'Family material difficulties Age 7'
With respect to the midlife part of the model:

1. It is unclear why and how the age 50 measures made. This is due in part to the fact that the tables describing the measures in the appendices are challenging to follow. A lot of descriptive information is repeated (e.g., labels, scale anchors) that could be presented more concisely. But, I am particularly concerned by the fact that social functioning and emotional functioning were combined into one "Emotional Well-Being" variable. What is the rationale for this? Typically, social functioning (e.g., social support network size, quality, etc.) is conceptualized as a protective factor that supports well-being, both physical and emotional. I think these should be included as separate constructs in the model. Additionally, the measure of quality of life appear to be more of a subjective well-being measure, akin to the Ryff Scales. Given its strong relationship with the emotional well-being measure, I question whether both should be included in the model or if one is a sufficient indicator of this aspect of well-being at age 50. Here I refer the authors to the work of L. Carstensen and S. Charles, and C. Ryff.

The age 50 measures (SF-36 and CASP-12) were utilised in this paper because we were fortunate enough to have them recorded in the same longitudinal dataset as our early-life predictors and mid-life mediators. We accept that social and emotional functioning can be treated separately within the SF-36 scale, but given the word constraints of this paper we preferred to follow the example of those such as Doll et al (2012) who separate the components of the SF-36 scale into the 'Physical/Emotional' duality.

Our self-reported measure of quality of life (CASP-12) is a theoretically informed measure of subjective well-being which closely follows the approach adopted by Ed Diener and his colleagues (Diener & Suh, 1997, Wiggins et al, 2008). The CASP measure (either 12- or 19-item versions) have been adopted in the US Health and Retirement Study (HRS), the Survey on Health, Ageing and

Retirement in Europe (SHARE) and the English Longitudinal Study of Ageing (ELSA). Please see www.casp.com for more instances of use.

CASP12 is indeed a subjective well-being measure, and bears a superficial resemblance to the Ryff scale, but it has more acceptance in a European context and has the benefit of exploring the four elements of quality of life at age 50 (control, autonomy, self-realisation and pleasure).

Minor points that require clarification:

1. In the SEM model, it would be helpful if there was consistency in the labelling with respect to age. That is, label all variables with their age label, like age50cog. Or align all variables vertically by age and add labels above or below the figure.

We accept this observation, and have re-labelled the elements of the model to make it clear to which age the measures refer: for example 'Family material difficulties Age 7.'

2. As previously mentioned, the tables are difficult to follow and more precise labelling of columns and rows is needed. For example, it is unclear what is presented in the column labelled "coding." Is this the possible or actual range? Seems to be the actual range, and it should be labelled as such.

We have re-labelled the columns and rows of the tables in the interests of more precise nomenclature (e.g. Appendix 1, now renamed Appendix 2)

3. It is unclear what the social class categories represent other than 1 = low. Do they map on to income brackets or some other meaningful metric that would help the reader understand the ranking? The same point applies to the qualifications measure.

The categories refer to the Registrar General's (RG) social classification of occupations, well-known in the context of UK research (see Szreter 1984), where it is typically preferred to a measure that combines education and income. RG Social class categories do not neatly map onto income categories (Wiggins et al 2004). The qualifications measure is that used in most UK educational research papers, such as Dearden et al 2002. Both measures are ordered categories.

4. Throughout the paper, there is an overuse of acronyms without sufficient description/definition, including in the figure captions.

We have revised the paper thoroughly to ensure all acronyms are defined before subsequent use.

Reviewer: 2

Reviewer Name: Kaitlin P. Ward, MSW

1. In the abstract and the introduction, it is unclear what the predictors in the model are, and why the authors have focused on these specific predictors. The introduction should lead the reader to understand why you chose the specific predictors you did. The authors keep mentioning childhood SES in the introduction, leading the reader to assume this would be the main focus. However, it seems the authors are focusing on SES, birthweight, maternal smoking, and breastfeeding? Or, is social class the authors' main predictor variable and birthweight, maternal smoking, and breastfeeding are covariates? I find myself constantly re-reading the article to figure out what the main predictor is.

We recognise that in epidemiological research papers there is a widely-accepted preference for identifying one primary predictor or 'exposure.' However, in the spirit of those using SEM such as Baron & Malmberg (2019), we have chosen to begin with four distinct predictors, since our SEM analysis enables us to follow the separate direct and mediated effects of these, as well as the inter-relationships between them. Our response here overlaps with the one given to Reviewer 1's first point.

a. In relation to the above point, the abstract mentions "childhood circumstances" broadly, then mentions birth SES and teenage cognition. The introduction should lead the reader to understand why you chose what time points you did (i.e., early childhood or adolescence).

We have adapted the introduction to explain that the longitudinal study surveyed participants at birth and ages 7, 11, 16 etc., which is why we have data at precisely those time-points. We have also amended the title to make it clear that we are focusing upon specific ages (birth, middle childhood and teenage years, as in Alatupa et al 2011).

b. To be clear, the authors have centered the article—even the article title—around the idea that they are focusing on early childhood. From what I understand, the main predictor that the authors focus on is the father's social class at the child's birth. That's not early childhood. There is also birthweight, maternal smoking and breastfeeding, but this is not early childhood either. These are measures at the time of birth. Then, the authors have measures at 7, 11, and 16. This is not early childhood; this is middle childhood to adolescence. It seems to me that they authors do not even have a single measure in the "early childhood" period.

We have adapted the title to clarify that we look at birth characteristics and childhood measures during school years, rather than **early** childhood.

2. In the introduction, the authors mention various theories that they are actually not testing. For example, the authors are not testing cumulative advantage/disadvantage; they are not testing for compensation; they are simply testing mediation. Instead of spending time discussing theories that the authors cannot and do not test in their analyses, the authors should be presenting theories (or at least justification) or WHY they are testing what they're testing. Why SES? What theories about SES—and the mechanisms through which SES works to influence future outcomes—apply to your study (perhaps Dr. Pam Davis-Kean's work would be helpful? Also Dr. Vonnie McLoyd and Dr. Rand Conger).

Our introductory discussion has been substantially restructured so as to emphasise the life-course approach (Carr, 2019, Elder 1994), which builds upon a model of cumulative advantage/disadvantage (Dannefer, 2003). We recognise that SES incorporates a 'proxy' for parental beliefs and behaviours, and how these influences play out in shaping a child's academic achievements. We attribute the exploration of the process of these mechanisms to Davis-Keane (2005) in our revised manuscript.

a. In relation to the point above, and as another example: the authors spend an entire paragraph discussing the mediating mechanisms that they never actually test. i.e., They don't measure whether parents are facilitating social/cultural/educational advantage. The authors need to tell the readers WHY the mechanisms that they actually test in the paper are important.

We accept this argument, and have therefore removed discussion of mechanisms we do not test, whilst explaining why we test the mechanisms that we do. We envisage testing other mechanisms in future work along the lines suggested by Brim et al's MIDUS exploration (2004, tables 1 & 2), but in the current context we have mentioned this as a limitation of our paper.

3. As the introduction stands, I am left wondering what the purpose of this study is. Why is the link from childhood SES to behavior in childhood/adolescence to age 50 outcomes important? Why is the link between childhood SES to cognition in childhood/adolescence age 50 outcomes important? Why do we care about those four outcomes specifically? As this introduction stands, I am not convinced that the variables/relationships the authors tested provide any meaningful contributions, academic or otherwise.

The reason for testing these relationships is that they have not been tested previously employing a structural equation model approach using such a large and rich nationally-representative longitudinal dataset following the same participants throughout 50 years of their lives. We have stressed this rationale more explicitly in our revised introduction. The purpose of our study is to investigate the extent to which childhood disadvantage/advantage has consequences for later life, and our outcome measures have been used in various combinations by European researchers (e.g. Cooper et al 2011, 2014). Nevertheless these previous researchers have not used such a broad outcome base as our approach, which we feel will lay the foundations for future work.

4. Why do the authors find the need to mention they examined fifty pathways? This makes me extremely concerned about issues of multicollinearity, model identification, whether the authors

followed the n:q rule of SEM, whether the model was identified, and so forth. Additionally, why were fifty pathways necessary? The authors did not tell the reader in the introduction why it was important for them to test those pathways theoretically or practically.

We did test for model identification, and were well satisfied that the n:q rule was followed with a population as large as 8,024. The number of free parameters is 261, and a sample-size-to-parameters ratio of 10:1 is regarded as imperative (Jackson 2003). In our revised paper we have mentioned this explicitly. The reason for testing so many pathways was related to the point the reviewer made below, namely that we wanted to be sure we had included enough covariates. We had to count the exact number of pathways in order to employ the requisite Bonferroni correction ($p=.001$ instead of $p=.05$).

5. Related to the conceptualization issues above, the authors need to provide a very convincing argument for why they would suspect that these pathways work differently for men and women. If the authors ran a multiple-group analysis, I should have a full paragraph in the introduction giving me an overview of the literature that would suggest that these mechanisms work differently for men and women.

We stress in our introduction that our study is largely exploratory: SEM need not be regarded as only being about confirmatory factor analysis. There are a priori substantive and empirical reasons to examine gender differences (see Ylöstalo & Brunila, 2018 and Cooper et al, 2011). The fact that we found six paths which were significantly different for men and women vindicates our approach.

A Why would the authors hypothesize that the mediation effects would be stronger for men than women? There is very little available evidence in the literature to justify formulating such a hypothesis.

The work of Cooper et al (2011) provides empirical support for investigating gender difference, but again we stress that our approach was exploratory: we did not start with a specific hypothesis, but we feel vindicated by the fact that we showed some mediation effects were stronger for men than for women.

6. The authors need to be a lot clearer about what they're actually testing. The authors point to appendix 1, but I'm still left wondering why they chose the variables that they did. I'd like things to be very clear: "we wanted to know if cognitive ability, behavioral problems, and education in middle childhood and adolescence mediated the relationship between SES at birth and age-50 outcomes."

We have revised our text to make it more clear exactly what we were testing, albeit in an exploratory spirit, having been informed in our approach by the research of Cooper et al (2011, 2014), Mishra et al (2011) and Clouston et al (2013).

Methods/Results

1. The authors need to discuss their measures. I understand the authors have limited space, but they at least need to discuss the primary predictor variable(s) (which I'm still unsure as to what that is/they are). Separate sections that discuss the predictors, mediators, and outcomes would be great.

We refer back to our earlier point, that we chose to begin with four distinct predictors based on characteristics at birth, the SEM approach enabling us to follow the separate direct and mediated effects of these, as well as the inter-relationships between them. Space limits our discussion of the mediators and outcomes, but we feel these are adequately dealt with in the supplementary material and the cited reference works validating these measures. In retrospect SES can be seen as the primary variable, but the other indicators measure important dimensions of mother's behaviour. The attraction of SEM is that it does allow the analyst to explore the role of specific predictors upon mid-life outcomes.

2. The authors need to discuss what the makeup of the sample looks like. Is it a truly representative sample? Is it mostly upper SES? Is it racially/economically diverse?

In our revised paper we have spelt out more explicitly that the birth sample were indeed truly representative, comprising 98.5% of all births in GB in that specific week of 1958. It is as racially and

economically diverse as the GB population itself was in 1958. Of course, attrition caused by loss to follow-up biases the sample a little as they get older (for instance female cohort members are slightly more likely to remain in this longitudinal survey as time goes on than men). Nevertheless we satisfied ourselves that, comparing the social class distribution of everyone at birth (N=17,415) with the distribution of that same birth social class variable among our chosen sample who stayed in the survey consistently (N=8,024), the class proportions remain satisfactorily intact:

	Full birth survey N=17,415	%	Our sample N=8024	%
Class I	746	4.5	386	5.2
Class II	2133	13.0	1127	15.1
Class III	9981	60.6	4498	60.1
Class IV	1995	12.1	871	11.6
Class V	1616	9.8	603	8.1
Total (valid Sclass)	16471	100.0	7485	100.0
Insuff data on parental Sclass	944		539	
Total in survey	17415		8024	

3. The authors need to provide alphas for all the measures with multiple indicators/items in the method section.

We have provided alphas for all Likert-Scale measures in the revised paper: Rutter Ages 7, 11, 16 and the three age 50 outcomes based on the SF-36 and CASP scales.

4. The authors need to present results of their measurement models. I at least need to know that they indicators all loaded onto the latent variable and that they had sufficient fit.

All of the manifest variables did indeed load onto the respective latent variables with satisfactory fit. We have included details in the revised supplementary material. We tested each one of our measurement models, but space did not permit the display of all the results. These can be made available on request.

5. The authors need to mention how they made the model fit. Did they have to use modification indices? Did any of the indicators need to be correlated?

We did indeed use modification indices, which is why for example the correlations appear for our four outcomes as they do in figure 2 and Appendix 5 (now re-named Appendix 6). We have been more explicit about this in our revised text.

6. The authors need to briefly mention whether their analyses met statistical assumptions. Were there outliers, non-normality, multicollinearity issues? Did you transform any of your variables to meet assumptions?

All analyses were conducted under full information maximum likelihood estimation and assumed to be robust to departures from statistical assumptions [Kaplan, (2009), Chapter 5].

7. In this study, did the authors have to use sampling weights or account for neighborhood/city clustering? Was this a multilevel model?

The sample consists of all births in Great Britain (GB) during one week in April, 1958 and is therefore an unclustered geographical sample of births across GB, so sampling weights weren't necessary. This was not a multilevel model.

8. If the authors are using FIML, why did they restrict the sample to the specifications found on page 5 lines 9-16?

The specifications (study members having at least one genuine response for each of the four separate age 50 life domains, and participating in at least one of the three cognitive and socio-emotional assessments) were a pragmatic course of action taken to protect against any substantial loss of data having an impact on the use of FIML. To that extent the use of FIML conditions on our restriction. Clearly, relaxing the restrictions would lead to a sensitivity check on the modelling but it was not considered to be necessary for this paper. This is supported by our reply to point 2 above.

9. What was the amount of missing data on your primary predictors, mediators, and outcome measures?

The descriptive statistics for each of these measures can be found in the Supplementary Material. Appendix 2, column 6 gives the percentage of missing data for each item.

10. Did the authors use bootstrapping for the mediation analysis? If so, how many?

No bootstrapping was used for the mediation analysis.

11. Throughout the results section, are the authors reporting standardized results (betas)?

Yes, all path coefficients are standardized.

12. When the authors are describing the results for men and women, how did they determine what was “stronger” for each? Did the authors use the DIFFTEST function in Mplus? Was it simply testing for structural invariance? If it’s the latter, how did you test for structural invariance, and did the authors use the scaling correction factor as an adjustment for when Mplus uses MLR?

As described in Appendix 7 (formerly Appendix 6), in performing the multigroup analysis by sex, we labelled each of the 50 pathways m_1, m_2, \dots, m_{50} in the male model and f_1, \dots, f_{50} in the female, and set variables $diff_1, \dots, diff_{50}$ ($diff_1 = m_1 - f_1$ etc) We found six paths of the 50 that were statistically different. For model parsimony we constrained all the other 44 paths to be equal for men and women, allowing just the six to vary, then noted that the first three had the greatest differences.

13. Why were there so few control variables? How did you settle on the control variables that you did?

We attempted to test a model of the life course with a manageable number of variables, each of which may well have an influence on later-life outcomes. To regard some of the variables as ‘controls’ would possibly be misleading as they are interesting in their own right. Clearly, the analysis is at risk of omitting key variables, e.g. occupational histories.

14. Why are you focusing on fathers' social class as your main predictor? Most studies that examine SES utilize some sort of combination of mothers' attributes (e.g., maternal education and income). The authors would need to justify this focus on fathers, and mention this in the introduction as well.

A combined social class measure based on mother’s and father’s social class was not feasible. NCDS is the second-oldest national longitudinal study in the world: in 1958 only 38% of mothers were economically active, so 62% had no RG social class.

Discussion

1. In general, I’m left confused as to how this study contributes to what already exists in the literature; how it relates to theory; and what the reader should take away from this study. I’m not sure why the authors ran the analyses they did.

We were pioneering the application of SEM on a very large nationally-representative dataset to explore the combined effects of early-life influences on mid-life outcomes. We argue that a unique contribution of our work is directed towards a better understanding of the inter-relationships between four outcomes taken at the same age. The thrust of the paper is to explore the legacy of early-life (dis-

)advantage for mid-life. Other authors, for example Wood et al (2017) adopt a similar methodological approach but only consider a single outcome.

2. The language used throughout the discussion adds to the abovementioned confusion. What were we trying to “tease out?” What to the authors mean by physical and emotional health “matter?”

We have re-written the text to avoid phrases such as ‘tease out’ and ‘matter.’

3. The authors' analyses do not support causation; the authors should remove all references to causation.

The references to causality were exclusively in the context of other authors' claims in the literature (e.g. Richards & Sacker 2010). We have amended our discussion of those authors.

4. The authors fail to provide an adequate overview of the study's limitations in the discussion section.

We have re-drafted the discussion section accordingly. One limitation of our study is the slight bias caused by sample loss (differential non-response or attrition). Another limitation concerns the conceptual coverage of our model: we are embarking on developing a framework which provides a conceptual approach following Brim et al (2004) together with an appropriate modelling strategy which we can use as a platform to extend our empirical analyses in the future as our CMs age.

References:

ALATUPA, S, PULKKI-RÅBACK, L, HINTSANEN, M, MULLOLA, S, LIPSANEN, J and KELTIKANGAS-JÄRVINEN, L (2011). Childhood Disruptive Behaviour and School Performance across Comprehensive School: A Prospective Cohort Study. *Psychology* 2(6), 542-551.

BARON & MALMBERG (2019). A vicious or auspicious cycle: The reciprocal relation between harsh parental discipline and children's self-regulation. *European Journal of Developmental Psychology* 16(3) 302-317 (doi: 10.1080/17405629.2017.1399875)

BRIM, O, RYFF, G, KESSLER, C.D & RONALD, C (2004). How healthy are we? A National Study of Well-Being at Midlife. Chicago, IL: University of Chicago Press.

CABLE, N, BARTLEY, M, McMUNN, A and KELLY, Y (2012) Gender differences in the effect of breastfeeding on adult psychological well-being. *European Journal of Public Health*, 22(5), 653-658.

CARR, D. (2019). Early-life influences on later life well-being: innovations and explorations. *The Journals of Gerontology, Series B: Psychological Sciences and Social Sciences*, 74(5), 829-831.

CASPI, A, HENRY, B, MCGEE, R.O, MOFFITT, T.E & SILVA, P (1995) Temperamental Origins of Child and Adolescent Behavior Problems: From Age Three to Age Fifteen *Child Development* 66 (1), 55-68 doi: [10.1111/j.1467-8624.1995.tb00855.x](https://doi.org/10.1111/j.1467-8624.1995.tb00855.x)

CLOUSTON, S.A, BREWSTER, P, KUH, D, RICHARDS, M, COOPER, R, HARDY, R, RUBIN, M.S and HOFER, S M (2013). The Dynamic Relationship Between Physical Function and Cognition in Longitudinal Aging Cohorts. *Epidemiologic Reviews* 35(1), 33–50 doi: [10.1093/epirev](https://doi.org/10.1093/epirev)

COOPER, R, STAFFORD, M, HARDY, R, AIHIE SAYER, A, BEN-SHLOMO, Y, COOPER, C & the HALCYON STUDY TEAM (2014). Physical capability and subsequent positive mental wellbeing in older people: findings from five HALCYon cohorts. *Age* 36(1), 445-456. doi: [10.1007/s11357-013-9553-8](https://doi.org/10.1007/s11357-013-9553-8)

COOPER, R, HARDY, R, AIHIE SAYER, A, BEN-SHLOMO, Y, BIRNIE, K, COOPER, C, CRAIG, L, DEARY, I.J, DEMAKAKOS, P, GALLACHER, J, McNEILL, G, MARTIN, R.M, STARR, J.M, STEPTOE, A, KUH, D and the HALCYON STUDY TEAM. (2011) Age and Gender Differences in

Physical Capability Levels from Mid-Life Onwards: The Harmonisation and Meta-Analysis of Data from Eight UK Cohort Studies. *PLoS One*, 6(11), e27899.

DANNEFER, D. (2003). Cumulative advantage/disadvantage and the life course: cross-fertilizing age and social science theory. *The Journals of Gerontology, Series B: Psychological Sciences and Social Sciences*, 58, 327-337.

DAVIS-KEAN, P.E (2005) The Influence of Parent Education and Family Income on Child Achievement: The Indirect Role of Parental Expectations and the Home Environment. *Journal of Family Psychology*, 19(2), 294-304.

DEARDEN, L, McINTOSH, S, MYCK, M and VIGNOLES, A. (2002). The returns to academic and vocational qualifications in Britain. *Bulletin of Economic Research*, 54(2), 249-274.

DIENER, E AND SUH , E (1997) Measuring quality of life: Economic, social, and subjective indicators. Social indicators Research, 40 (1-2), 189-216.

DOLL, H.A, PETERSEN , S.E.K and STEWART-BROWN, S.L (2012) Obesity and Physical and Emotional Well-Being: Associations between Body Mass Index, Chronic Illness, and the Physical and Mental Components of the SF-36 Questionnaire. *Obesity* 8(2), 160-170 doi: [10.1038/oby.2000.17](https://doi.org/10.1038/oby.2000.17)

ELDER, G. (1994). Time, human agency, and social change: perspectives on the life course. *Social Psychology Quarterly*, 57, 4-15.

GALE, C. R, COOPER, R, CRAIG, L, ELLIOTT, J, KUH, D, RICHARDS, M & DEARY, I. J (2012). Cognitive function in childhood and lifetime cognitive change in relation to mental wellbeing in four cohorts of older people. *PLoS ONE*, 7(9), e44860. doi: [10.1371/journal.pone.0044860](https://doi.org/10.1371/journal.pone.0044860)

JACKSON, D,L (2003) Revisiting Sample Size and Number of Parameter Estimates: Some Support for the N:q Hypothesis. *Structural Equation Modeling* 10(1), 128-141.

KAPLAN, D (2009). Statistical Assumptions underlying SEM in Structural Equation Modelling: Foundations and Extensions, Chapter 5, 2nd Ed., Sage.

MARMOT, M. G., FUHRER, R., ETTNER, S.L., MARKS, N.F., BUMPASS, L.L. and RYFF, C.D (1998) Contribution of psychosocial factors to socioeconomic differences in health. *Millbank Quarterly* 76, 403-48.

MISHRA, G.D, GALE, C.R, AIHIE SAYER, A, COOPER, C, DENNISON, E. M, WHALLEY, L.J, CRAIG, L, KUH, D, DEARY, I. J and the HALCYON STUDY TEAM (2011) How useful are the SF-36 sub-scales in older people? Mokken scaling of data from the HALCYon programme. *Quality of Life Research*, 20 (7), 1005-1010. (doi:[10.1007/s11136-010-9838-7](https://doi.org/10.1007/s11136-010-9838-7)). (PMID:21225350)

PARSONS, S, GREEN, F, PLOUBIDIS, G.B, SULLIVAN, A and WIGGINS, R.D (2017) The influence of private primary schooling on children's learning: Evidence from three generations of children living in the UK. *British Educational Research Journal* 43(5), 823-847 (doi: [10.1002/berj.3300](https://doi.org/10.1002/berj.3300))

POWER, C and MATTHEWS, S. (1997) Origins of health inequalities in a national population sample. *Lancet*, 350(9091), 1584-89.

SACKER, A, KELLY, Y, IACOVOU, M, CABLE, N and BARTLEY, M (2013) Breast feeding and intergenerational social mobility: what are the mechanisms? *Archives of Disease in Childhood*, 98(9), 666-671.

SZRETER, S.R.S (1984) The Genesis of the Registrar-General's Social Classification of Occupations. *British Journal of Sociology* 35(4), 522-54

WIGGINS, R.D, SCHOFIELD, P, BARTLEY, M, SACKER, A and HEAD, J (2004). Social Determinants of changes in minor psychiatric morbidity over time in the British Household Panel Study, 1991-1998. *Journal of Epidemiology and Community Health*, Vol. 58 pp.779-791.

WIGGINS, R.D, NETUVELI, G, HYDE, M, HIGGS,P and BLANE, D (2008). The evaluation of a self-enumerated scale of quality of life (CASP-19) in the context of research on ageing: a combination of exploratory and confirmatory approaches. *Social Indicators Research* 89(1), 61-77.

WOOD, N, BANN, D, HARDY, R, GALE, C, GOODMAN, A, CRAWFORD, C and STAFFORD, M (2017) Childhood socioeconomic position and adult mental wellbeing: evidence from British birth cohort studies. *PLoS One*, 10, e0185798.

YLÖSTALO, H and BRUNILA, K. (2018) Exploring the possibilities of gender equality pedagogy in an era of marketization. *Gender and Education* 30(7), 917-933.

VERSION 2 – REVIEW

REVIEWER	Erin Barker Concordia University Montreal
REVIEW RETURNED	26-Aug-2019

GENERAL COMMENTS	The authors responded well to the two reviewers who agreed that the goals of the study needed to be clarified. The restructuring of and more direct and precise language in the Introduction has helped to clarify the goals. With that, I no longer think that Figure 1 is needed and I actually think it detracts from the presentation of the goals of the study. For example, in the conceptual model “birth social circumstances” are presented together as one domain of interest. This brings me back to my original question as to why each indicator was included as a manifest variable rather than a latent construct. The authors responded that it was important to separate the indicators. Having them presented together in the conceptual model confuses things. The revised introduction along with the SEM figure is sufficient and much clearer because the nuance that the authors highlight as the main contributions is shown at each age in the full model. If the authors choose to retain Figure 1 a few edits are needed. The labels need to be edited to be more consistent with Figure 2. Birth social circumstances should be separated from birth health indicators. Ages should be added at all levels and/or all variables should be aligned by age. Family difficulties should be called family material difficulties. Figure 2 is improved with the addition of the age markers, but the Age 42 notation is missing from the Qualls box. Appendix 2 is also improved, but Appendix 3 still requires work. Only one column in the table is needed. This 4-page table could be condensed into a single 1-2 page table that is much easier to read. There is no need to repeat the same information in two columns or repeat the ratings scale within groups of items that use the same scale or repeat the domain name. I suggest the following format where all of the physical health domain questions are grouped in the first half of the table and all of the emotion items in the second half. In Appendix 5, please add age notation for all variables. Some have it and some do not. Please be consistent. Finally, with the removal of Figure 1 and edits to Appendix 3, there should be room to include information about the measurement model as requested by the other reviewer. Suggested Appendix 3 Formatting SF -36 Item Number, Response Options, and Wording by Model
---

	Domain Physical Health Domain Self -assessed General Health 1. In general, would you say your health is: (1) Excellent; (2) Very good; (3) Good; (4) Fair; (5) Poor Self -assessed General Health compared to one year ago 2. Compared to one year ago, how would you rate your health in general now? (1) Much better now than one year ago; (2) Somewhat better now than 1 year ago; (3) About the same; (4) Somewhat worse now than 1 year ago; (5) Much worse now than one year ago Health now limiting specific physical activities: The following items are about activities you might do during a typical day. Does your health now limit you in these activities? If so, how much? (1) Yes, limited a lot; (2) Yes, limited a little; (3) No, not limited at all 3. Vigorous activities, such as running, lifting heavy objects, participating in strenuous sports 4. Moderate activities, such as moving a table, pushing a vacuum cleaner, bowling, or playing golf? 5. Lifting or carrying groceries? 5. Climbing several flights of stairs? 6. Climbing one flight of stairs? 7. Bending, kneeling, or stooping? 8. Walking more than a mile? 9. Walking several blocks? 10. Walking one block? 11. Bathing or dressing yourself? 21. How much bodily pain have you had during the past 4 weeks? (1) None; (2) Very mild; (3) Mild; (4) Moderate; (5) Severe; (6) Very severe Emotional Well-Being Domain These questions are about how you feel and how things have been with you during For each question, please give the one answer that comes closest to the way you have been feeling. How much of the time during the past 4 weeks... (1) All of the time; (2) Most of the time; (3) A good bit of the time; (4) Some of the time; (5) A little of the time; (6) None of the time 23. Did you feel full of pep? 24. Have you been a very nervous person? 25. Have you felt so down in the dumps that nothing could cheer you up? 26. Have you felt calm and peaceful? 27. Did you have a lot of energy? 28. Have you felt downhearted and blue? 29. Did you feel worn out? 30. Have you been a happy person? 31. Did you feel tired? etc.
--	---

REVIEWER	Kaitlin P. Ward, MSW University of Michigan, USA
REVIEW RETURNED	25-Aug-2019

GENERAL COMMENTS	The authors responded well to most of the concerns that were raised. In particular, the introduction and limitations sections are much improved. I am still left unsure whether the statistical analyses as scientifically sound (e.g., whether the data met statistical assumptions, misspecifications to the model, poor model fit, lack of bootstrapping). I feel these concerns would need to be addressed in
---

	order for me to recommend this manuscript for publication.  1. In the Abstract under Objectives, change “early childhood” to “middle childhood.” 2. Even based on the previous research that the authors cite for examining gender differences (e.g., Ylöstalo & Brunila, 2018 and Cooper et al, 2011), these would not provide substantial evidence for the authors to hypothesize beforehand that there would be “stronger mediated links for men than women.” I recommend that the authors use more exploratory language in the Abstract under Objectives, and replace it with something similar to the following: “We also explore whether the effects of childhood circumstances on mid-life physical and emotional well-being differ between men and women.” 3. Thank you for changing the title to be a more accurate reflection of the study. However, the revised title is too long. I recommend shortening the title. 4. FIML can help will small departures from statistical assumptions, but this does not negate the need to inform the reader whether or not your data met statistical assumptions. If the data have multicollinearity problems, are extremely skewed, have outliers, and so forth, FIML will not be able to handle these departures. Please include whether your data met the statistical assumptions for the models you chose to run. 5. A long line of research suggests that bootstrapping provides the most robust testing of mediation effects (see David Kenny and David MacKinnon’s work). Most authors recommend that researchers use at least 10,000 bootstraps. At the minimum, I recommend that the authors run 1,000 bootstraps. 6. In your results section, add a Beta to all your results so that it is clear to the reader that your results are standardized. For example, “The cognitive variable, in turn, has a strong direct effect on age 11 cognition ($\beta = .92$).” 7. The model fit is not sufficient, suggesting that the model is misspecified. A model with a CFI of .88 and a TLI of .86 is concerning. The model may be missing covariance estimates or other pathways; given that the authors utilized modification indices and estimated so many pathways, the poor fit of the model is very concerning. The authors need to reexamine the model to determine why their model fits poorly. From a quick look at the figures, I believe the authors need to have covariance estimates between the Rutter and Cognitive measures that occur at the same ages (e.g., Rutter age 7 and cognitive Age 7; Rutter age 11 and cognitive age 11, and Rutter Age 16 and cognitive age 16) instead of regressing the cognitive measure onto the Rutter. 8. In Appendix 7, remove the wording “male p-value falls just sort of satisfying” mean? Simply state that, when taking the Bonferroni correction into account, the male estimate is non-significant. Also take out the wording “...female p-value is nowhere near.” Simply state that the female estimate is non-significant. 9. Specify in all your figures whether the numbers presented are standardized or unstandardized.
--	---

REVIEWER	Katerina Marcoulides University of Minnesota
REVIEW RETURNED	25-Sep-2019

GENERAL COMMENTS	After reviewing the paper “A Longitudinal Analysis of the 1958 British
--

	Birth Cohort Study examining the extent to which cognitive ability, behavioural problems and education in middle childhood and adolescence mediate the relationship between circumstances at birth and well-being outcomes at age 50 years”, I believe that the paper is interesting and, following some revisions (discussed further below) has the potential to be a strong contribution to the literature. That being said, I am unsure whether this paper is in line with the aims and scope of this journal. Another major concern I have with the paper is that the aim or purpose of the paper is not entirely clear. Based on the title of the paper, it leads the reader to believe that there is a specific idea about the relationships among variables that is being specifically examined. However, upon further reading, it seems as though the analyses are more exploratory in nature. Related to this issue, the objectives mentioned in the abstract are not clearly stated in the actual paper itself. The objectives in the abstract indicate that the authors “... aim to examine the relative contributions of pathways from early childhood/adolescence to mid-life well-being, health and cognition, in the context of family socio-economic status (SES) at birth, educational achievement and early-adulthood SES.” The authors hypothesize “... that effects of childhood circumstances on mid-life physical & emotional well-being are strongly mediated by cognitive performance during school years, with stronger mediated links for men than women from childhood-to-adult SES.” However, this hypothesis is not clearly described in the body of the paper. The authors need to provide a stronger foundation in the existing literature to then propose this hypothesis. Additionally, the authors need to be more upfront about this specific hypothesis, how this hypothesis was tested in their structural equation model, and what path(s) were not supported relative to this hypothesis. The authors are forthcoming about their use of modification indices, but without their initial hypothesis being clearly stated in the body of the paper, it is impossible to know which paths resulted from suggestions of the modification indices (and are thus more exploratory in nature). The authors did a good job of alleviating any concerns related to attrition and the large number of pathways in their SEM. The authors refer to the inter-relationships between the latent variables in Figure 1, however, based on the diagram, there are no latent variables, only observed variables. Latent variables should be denoted by circles/ellipses (observed variables are denoted by squares/rectangles) as was done in Figure 2. Finally, the authors state that “there is a strong argument for combining these outcomes in order to provide a holistic perspective”, however, this statement is unclear and potentially misleading. The authors aren’t actually combining the outcomes, there are still four separate outcomes. Perhaps they mean say combining these outcome in a single model. In the same sentence, the authors also seem to indicate that examining gender differences is somehow related to the strong argument for combining these outcomes. It is unclear how these two ideas are related.
--	---

VERSION 2 – AUTHOR RESPONSE

Reviewer: 2
Reviewer Name: Kaitlin P. Ward, MSW
Institution and Country: University of Michigan, USA

Please state any competing interests or state 'None declared': None declared.

Please leave your comments for the authors below

The authors responded well to most of the concerns that were raised. In particular, the introduction and limitations sections are much improved. I am still left unsure whether the statistical analyses are scientifically sound (e.g., whether the data met statistical assumptions, misspecifications to the model, poor model fit, lack of bootstrapping). I feel these concerns would need to be addressed in order for me to recommend this manuscript for publication.

1. In the Abstract under Objectives, change “early childhood” to “middle childhood.”

This has been duly changed, as recommended by the reviewer.

2. Even based on the previous research that the authors cite for examining gender differences (e.g., Ylöstalo & Brunila, 2018 and Cooper et al, 2011), these would not provide substantial evidence for the authors to hypothesize beforehand that there would be “stronger mediated links for men than women.” I recommend that the authors use more exploratory language in the Abstract under Objectives, and replace it with something similar to the following: “We also explore whether the effects of childhood circumstances on mid-life physical and emotional well-being differ between men and women.”

Our introduction and abstract have been altered to place more emphasis on the exploratory nature of our research as opposed to the testing of specific hypotheses. In particular, regarding gender differences we have adopted the wording thoughtfully suggested by the reviewer.

3. Thank you for changing the title to be a more accurate reflection of the study. However, the revised title is too long. I recommend shortening the title.

We have shortened the title from 40 to 24 words, with the emphasis altered a little (as above) to stress the paper's exploratory approach.

4. FIML can help with small departures from statistical assumptions, but this does not negate the need to inform the reader whether or not your data met statistical assumptions. If the data have multicollinearity problems, are extremely skewed, have outliers, and so forth, FIML will not be able to handle these departures. Please include whether your data met the statistical assumptions for the models you chose to run.

FIML estimation in MPlus (MLR) is robust for handling departures from symmetry and non-normality. Nevertheless we accept the need to address the issues of outliers and multicollinearity.

To test for the effect of outliers, we ran a multivariate analysis of Mahalanobis distance on our thirteen predictors and mediators, comparing this with a Chi-Square distribution with thirteen degrees of freedom, using the method of T.L Grande (2015) cited in two subsequent doctoral theses (Bakthavatchalam 2019 and Weaver 2019).

We found 96 outliers at the $p=.001$ level among our population of 8,024. We re-ran the model excluding these cases with a reduced sample of 7,928 for each of the four outcomes.

The model fit indices remained unchanged, and for each estimated path the β values were identical to within two decimal places. The results were not affected by exclusion of outliers. Detailed results are available on request.

Checking for multicollinearity, we ran correlations between all our exposures and mediators, and found no level greater than 0.49 apart from our 'repeated measures' Age7Cog and Age11Cog, which understandably correlate highly with Age16cog. As a test of any adverse effect, we ran the same model omitting both Age7Cog and Age11Cog, and found that the estimates with regard to our outcome variables remained the same. Nevertheless, we feel the model remains stronger by retaining Age7Cog and Age11Cog, so as to note the associations with family material difficulties and Rutter scores at the respective ages.

5. A long line of research suggests that bootstrapping provides the most robust testing of mediation effects (see David Kenny and David MacKinnon's work). Most authors recommend that

researchers use at least 10,000 bootstraps. At the minimum, I recommend that the authors run 1,000 bootstraps.

We ran 1,000 bootstraps with ML estimation in MPlus. The standard errors and their ratios to point estimates were very similar indeed to those produced by MLR. Detailed results are available on request.

6. In your results section, add a Beta to all your results so that it is clear to the reader that your results are standardized. For example, “The cognitive variable, in turn, has a strong direct effect on age 11 cognition ($\beta = .92$).”

We have inserted Betas throughout, to underline the fact that all our results are standardised.

7. The model fit is not sufficient, suggesting that the model is misspecified. A model with a CFI of .88 and a TLI of .86 is concerning. The model may be missing covariance estimates or other pathways; given that the authors utilized modification indices and estimated so many pathways, the poor fit of the model is very concerning. The authors need to reexamine the model to determine why their model fits poorly. From a quick look at the figures, I believe the authors need to have covariance estimates between the Rutter and Cognitive measures that occur at the same ages (e.g., Rutter age 7 and cognitive Age 7; Rutter age 11 and cognitive age 11, and Rutter Age 16 and cognitive age 16) instead of regressing the cognitive measure onto the Rutter.

We re-ran the model with covariance estimates between the Rutter and cognitive measures at the same ages instead of regressing the cognitive measure onto the Rutter (see revised Fig 2, now re-labelled Fig.1). We agree that the inclusion of bi-directional influences makes better sense than in our original formulation, so we have revised all the reported results to reflect the model in this form. However, though we prefer this model formulation for the paper, the effect on the model fit was minimal.

We are of the opinion that the criteria adopted for assessing model fit are adequate, given that our approach is broadly exploratory. Our RMSEA is very satisfactory at 0.045, and our CFI and TLI values (.88 and .86) closely approach conventionally-acceptable levels. See for example, Grove, R, Hoekstra, R.A et al (2018), whose RMSEA is less favourable, with CFI & TFI of 0.9; Kiernan, K.E and Huerta, M.C. (2008) also with CFI & TFI of 0.9; or Hysing et al 2016, whose SEM has virtually the same CFI and TLI as our model.

8. In Appendix 7, remove the wording “male p-value falls just sort of satisfying” mean? Simply state that, when taking the Bonferroni correction into account, the male estimate is non-significant. Also take out the wording “...female p-value is nowhere near.” Simply state that the female estimate is non-significant.

We fully accept this point, and have amended the text accordingly.

9. Specify in all your figures whether the numbers presented are standardized or unstandardized.

As noted in point 6, we have added a Beta to all our results, since there are none which are unstandardised.

Reviewer: 1

Reviewer Name: Erin Barker

Institution and Country:

Concordia University

Montreal

Please state any competing interests or state ‘None declared’: none declared

Please leave your comments for the authors below

The authors responded well to the two reviewers who agreed that the goals of the study needed to be clarified. The restructuring of and more direct and precise language in the Introduction has helped to clarify the goals. With that, I no longer think that Figure 1 is needed and I actually think it detracts from the presentation of the goals of the study. For example, in the conceptual model “birth social

circumstances” are presented together as one domain of interest. This brings me back to my original question as to why each indicator was included as a manifest variable rather than a latent construct. The authors responded that it was important to separate the indicators. Having them presented together in the conceptual model confuses things. The revised introduction along with the SEM figure is sufficient and much clearer because the nuance that the authors highlight as the main contributions is shown at each age in the full model. If the authors choose to retain Figure 1 a few edits are needed. The labels need to be edited to be more consistent with Figure 2. Birth social circumstances could be separated from birth health indicators. Ages should be added at all levels and/or all variables should be aligned by age. Family difficulties should be called family material difficulties.

We fully accept this point, and have accordingly omitted Figure 1 in the revised text.

Figure 2 is improved with the addition of the age markers, but the Age 42 notation is missing from the Quals box.

This has been duly rectified.

Appendix 2 is also improved, but Appendix 3 still requires work. Only one column in the table is needed. This 4-page table could be condensed into a single 1-2 page table that is much easier to read. There is no need to repeat the same information in two columns or repeat the ratings scale within groups of items that use the same scale or repeat the domain name. I suggest the following format where all of the physical health domain questions are grouped in the first half of the table and all of the emotion items in the second half.

Appendix 3 has been revised as suggested.

In Appendix 5, please add age notation for all variables. Some have it and some do not. Please be consistent.

Appendix 5 has been revised as suggested.

Finally, with the removal of Figure 1 and edits to Appendix 3, there should be room to include information about the measurement model as requested by the other reviewer.

As requested, an Appendix 5 has been introduced, with full details of the measurement models, which we had tested before beginning our full SEM.

Reviewer: 3

Reviewer Name: Katerina Marcoulides

Institution and Country: University of Minnesota

Please state any competing interests or state 'None declared': None declared

Please leave your comments for the authors below

After reviewing the paper “A Longitudinal Analysis of the 1958 British Birth Cohort Study examining the extent to which cognitive ability, behavioural problems and education in middle childhood and adolescence mediate the relationship between circumstances at birth and well-being outcomes at age 50 years”, I believe that the paper is interesting and, following some revisions (discussed further below) has the potential to be a strong contribution to the literature. That being said, I am unsure whether this paper is in line with the aims and scope of this journal.

We would contend that the paper is well in line with the journal's aims and scope. The journal's website states:

“We consider papers addressing research questions in clinical medicine, public health and epidemiology. ... Our aim is to provide a home for all properly conducted medical research to be fully reported, after a rigorous and transparent peer review process.”

Our paper fits well within the discipline of epidemiology and public health: indeed we consider it to be of a similar methodological and analytical approach to Richards et al (BMJ Open, 2019).

Another major concern I have with the paper is that the aim or purpose of the paper is not entirely clear. Based on the title of the paper, it leads the reader to believe that there is a specific idea about the relationships among variables that is being specifically examined. However, upon further reading, it seems as though the analyses are more exploratory in nature.

Related to this issue, the objectives mentioned in the abstract are not clearly stated in the actual paper itself. The objectives in the abstract indicate that the authors "... aim to examine the relative contributions of pathways from early childhood/adolescence to mid- life well-being, health and cognition, in the context of family socio-economic status (SES) at birth, educational achievement and early-adulthood SES." The authors hypothesize "... that effects of childhood circumstances on mid-life physical & emotional well-being are strongly mediated by cognitive performance during school years, with stronger mediated links for men than women from childhood-to-adult SES." However, this hypothesis is not clearly described in the body of the paper.

We accept this observation, and (with reference to our answer to Reviewer 2 points 2 & 3 above), we hope that in our revised paper, our emphasis on the exploratory nature of the research reduces the reader's expectation of a hypothesis being tested, hopefully addressing this concern.

The authors need to provide a stronger foundation in the existing literature to then propose this hypothesis. Additionally, the authors need to be more upfront about this specific hypothesis, how this hypothesis was tested in their structural equation model, and what path(s) were not supported relative to this hypothesis. The authors are forthcoming about their use of modification indices, but without their initial hypothesis being clearly stated in the body of the paper, it is impossible to know which paths resulted from suggestions of the modification indices (and are thus more exploratory in nature).

Again, we hope that our emphasis on the exploratory nature of the research in our revised paper addresses this concern. The main paths which resulted from modification indices were the correlations between our Age 50 outcome measures.

The authors did a good job of alleviating any concerns related to attrition and the large number of pathways in their SEM.

The authors refer to the inter-relationships between the latent variables in Figure 1, however, based on the diagram, there are no latent variables, only observed variables. Latent variables should be denoted by circles/ellipses (observed variables are denoted by squares/rectangles) as was done in Figure 2.

With reference to our answer to Reviewer 1 point 1, we have omitted Figure 1 from our revised paper, and hope this fully addresses this concern.

Finally, the authors state that "there is a strong argument for combining these outcomes in order to provide a holistic perspective", however, this statement is unclear and potentially misleading. The authors aren't actually combining the outcomes, there are still four separate outcomes. Perhaps they mean say combining these outcome in a single model. In the same sentence, the authors also seem to indicate that examining gender differences is somehow related to the strong argument for combining these outcomes. It is unclear how these two ideas are related.

We accept that this statement arguing for combining the four outcomes is rather misleading. We have omitted the statement from the revised version.

References

Grove, R, Hoekstra, R.A et al (2018) Special interests and subjective wellbeing in autistic adults Autism Research 11 (5), 766-755

Hysing, M, Sivertsen, B, Garthus-Niegel, S & Eberhard-Gran, M (2016) Pediatric sleep problems and social-emotional problems. A population-based study. Infant Behavior and Development Volume 42 (Feb 2016) 111-118

Kiernan, K.E and Huerta, M.C. (2008) Economic Deprivation, Maternal Depression, Parenting and Children's Cognitive and Emotional Development in Early Childhood. *British Journal of Sociology*, 59(4), 783–806.

Grande, T.L (2015) Identifying Multivariate Outliers with Mahalanobis Distance in SPSS. Youtube.

Richards, M, James, S.N, Sizer, A, Sharma,N, Rawle, M.J, Davis, D and Kuh, D (2019) Identifying the lifetime cognitive and socioeconomic antecedents of cognitive state: seven decades of follow-up in a British birth cohort study. *BMJ Open* 9(4), e024404.

Weaver, J.L (2019) Student-Professor Interaction, Academic Integration, Self-Efficacy, and Persistence in Nontraditional Students. Doctoral Thesis, Grand Canyon University, 13900983.

Bakthavatchalam, V.P (2019) Motivation to Conduct Research in a Rapidly Evolving Academic Environment: A Study of Coimbatore's Engineering Institutions. Doctoral Thesis, University of Plymouth

VERSION 3 – REVIEW

REVIEWER	Erin Barker Concordia University Canada
REVIEW RETURNED	18-Nov-2019

GENERAL COMMENTS	I have no additional comments for the authors.
--

REVIEWER	Katerina M. Marcoulides University of Minnesota, USA
REVIEW RETURNED	11-Nov-2019

GENERAL COMMENTS	The authors have clarified a number of issues raised with the previous version of the manuscript. The following aspects that need to be addressed remain.  Figure 1 provides the model fit criteria for the full model, but these do not appear to meet the criteria for model fit specified on page 5, lines 10-13. The CFI is 0.875 and the Tucker-Lewis Index is 0.863, both of which do not meet the defined criteria of good fit. Similarly questionable model fit criteria are also provided in Appendix 8. Appendix 5 provides the separate measurement models for the eight latent variables in the SEM analysis. However, some of the indicator loadings are either extremely low or fairly low in value and suggest that these indicators do not really measure the specified factor well. For example, the factor loadings for “Copying designs” in Age11Cog is 0.12. Similarly, “Letter cancellation speed” in Age50Cog is 0.12, and there are many more such low factor loadings provided. On page 6, line 36, a beta coefficient is provided for the direct effect of social class on educational/vocational qualifications (the same notation is also used throughout to denote other path coefficients). Is this notation meant to be representative of a standardized coefficient or an unstandardized coefficient? Normally in SEM notation, the direct effect of a predictor on an outcome is denoted by a gamma.
--

4. Finally, in Appendix 5 the indicator loadings in SEM notation are normally denoted by a lower case lambda and not a beta.
--

VERSION 3 – AUTHOR RESPONSE

In response to point 1:

We have amended the wording on page 5, lines 10-13, to emphasise, in support of our approach, that in many published SEM papers, each of these criteria are regarded as 'indicative' and are not always strictly adhered to individually as arbiters of model validity. We provide a recent reference (Lai & Green 2016)* whose authors argue that it is not necessary to disregard the model because an index fails to meet the cut-off criterion, as substantive contributions may outweigh the diagnostics. Lai & Green have been cited by 87 published papers.

In response to point 2:

While we recognise that the factor loadings for Copying Designs (within latent variable Age11Cog), Letter Cancellation Speed (within Age50Cog) and Social Functioning (within EWB) are low at 0.12, we have tried running the full SEM removing these, and find this does not improve the overall fit of the model substantially (CFI=0.878, TLI=0.865). We also observed that the correlation of each of the three latent variables in turn with and without these indicators is between 0.99 and 1. We therefore prefer to keep these factors in the model, particularly as Social Functioning is a well-validated component of the SF-36 sub-domains of life.

In response to point 3:

We have amended the results on p.6, inserting a note clarifying that all reported coefficients are standardised. We have taken out the Beta notation to avoid confusion.

In response to point 4:

As above, we have similarly amended Appendix 5 to take out the beta notation.

* Lai, K & Green, S.B (2016) The problem with having two watches: Assessment of fit when RMSEA and CFI disagree. *Multivariate Behavioral Research* 51 (2-3), 220-239